# Objects Matter: Object-Centric World Models Improve Reinforcement Learning in Visually Complex Environments

**Weipu Zhang, Adam Jelley, Trevor McInroe, Amos Storkey**

weipuzhang.academic@gmail.com,
{adam.jelley, t.mcinroe, a.storkey}@ed.ac.uk

University of Edinburgh

## Abstract

Deep reinforcement learning has achieved remarkable success in learning control policies from pixels across a wide range of tasks, yet its application remains hindered by low sample efficiency, requiring significantly more environment interactions than humans to reach comparable performance. Model-based reinforcement learning (MBRL) offers a solution by leveraging learnt world models to generate simulated experience, thereby improving sample efficiency. In visually complex environments, small or dynamic elements can be critical for decision-making. However, traditional MBRL methods in pixel-based environments typically rely on auto-encoding with an $L_2$ loss, which is dominated by large areas and often fails to capture decision-relevant details. To address these limitations, we propose an **object-centric MBRL pipeline**, which integrates recent advances in computer vision to allow agents to focus on key decision-related elements. We demonstrate OC-STORM's practical value in overcoming the limitations of conventional MBRL approaches on both Atari games and the visually complex game Hollow Knight.

## 1 Introduction

Over the past decade, deep reinforcement learning (DRL) algorithms have demonstrated remarkable capabilities across a wide-range of tasks (Silver et al., 2016; Mnih et al., 2015; Hafner et al., 2023). However, applying DRL to real-world scenarios remains challenging due to low sample efficiency, meaning DRL agents require significantly more environment interactions than humans to achieve comparable performance. A promising solution to this problem is model-based reinforcement learning (MBRL) (Sutton & Barto, 2018; Ha & Schmidhuber, 2018). By utilizing predictions from a learned world model, MBRL enables agents to generate and learn from simulated trajectories, thereby reducing reliance on direct interactions with the real environment and improving sample efficiency.

Recent MBRL methods typically train the world model in a self-supervised autoregressive manner (Hafner et al., 2023; Zhang et al., 2023; Micheli et al., 2023; Alonso et al., 2024). The training objective in pixel-based environments is usually defined with $L_2$ or Huber (Huber, 1964) reconstruction loss. While such an approach is simple and effective in many cases, it can fail to capture decision-relevant information. When the key targets are too small, the background is dynamic, or there are too many decision-irrelevant objects in the scene, the agent can easily miss these key targets, leading to poor performance.

Meanwhile, recent advances in computer vision, such as open-set detection and segmentation technologies including SAM (Kirillov et al., 2023; Ravi et al., 2024), Cutie (Cheng et al., 2023), and GroundingDINO (Liu et al., 2023), have revolutionized our ability to identify objects in diverse

environments. These models excel at detecting or segmenting objects in out-of-domain cases without further finetuning. By integrating these capabilities into reinforcement learning, the agent can immediately focus on essential decision-relevant elements, bypassing the need to study how to extract key information from raw observations.

To mitigate the limitations of reconstruction losses in previous MBRL methods, we propose an **object-centric model-based reinforcement learning pipeline** that leverages these advances in computer vision. This pipeline involves four steps:

1. Annotating key objects in a small number of frames using segmentation masks.
2. Extracting object features through a parameter-frozen pre-trained vision foundation model conditioned on these annotations. In this work, we use Cutie (Cheng et al., 2023).
3. Utilizing both these object features and the raw observations as inputs for training an object-centric world model that predicts the dynamics of the environment while considering the relationships between different objects and the scene.
4. Training the policy with imagined trajectories generated by the world model.

Since the MBRL component of this pipeline is based on STORM (Zhang et al., 2023), we name our method **OC-STORM**. To our knowledge, we are the first to successfully adopt object-centric learning on Atari and the visually more complex game Hollow Knight without relying on an extensive number of labels or accessing internal game states (Delfosse et al., 2023; Jain, 2024). OC-STORM outperforms the baseline STORM on 18 of 26 tasks in the Atari 100k benchmark and achieves the best-known sample efficiency on several Hollow Knight bosses.

## 2 Preliminaries

### 2.1 Object Extraction

Object detection and segmentation have been active areas of research in recent years, leading to the development of various influential methods. Appendix B.3 provides a brief review of the methods which we considered for extracting object representations for reinforcement learning agents (Cheng & Schwing, 2022; Kirillov et al., 2023; Zhang et al., 2024; Ravi et al., 2024; Redmon et al., 2016; Jocher et al., 2023; Liu et al., 2023; Locatello et al., 2020; Kipf et al., 2022; Elsayed et al., 2022; Wang et al., 2023; Xu et al., 2023). After careful consideration of these potential methods, we selected Cutie (Cheng et al., 2023) as the object feature extractor.

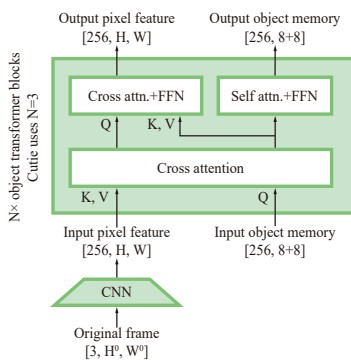

Figure 1: A simplified illustration of the object transformer in Cutie. The tuples in square brackets represent the shapes of the corresponding tensors. For more details, please refer to the original paper (Cheng et al., 2023).

Cutie is a retrieval-based video object segmentation algorithm capable of generating consistent representations across frames. The core component of Cutie is the object transformer, depicted in Figure 1, which integrates pixel-level and object-level features. This integration enriches pixel-level information with high-level object semantics, thereby improving segmentation accuracy. The object-level feature is a compact vector, which we employ to represent the corresponding object.

The object memory of Cutie is first initialized with mask-pooling over pixel features and then refined with the object transformer. For the pooling mask, Cutie generates 16 different masks to cover different aspects of the object. The 16 object memory features and pooling masks are equally divided into foreground and background components by default. The first half integrates information belonging to the object, while the second half is targeted toward the background. Since backgrounds may vary across different scenes, the background features can shift and become inconsistent. Therefore, only

the first 8 foreground features are used as input to the agent. Consequently, each object is represented by a $256 \times 8 = 2048$-dimensional vector.

As the pixel features are combined with positional embeddings, the resulting object memory encapsulates both the state and position of objects. Therefore, if the object is segmented correctly, this representation should theoretically be sufficient for decision-making. Evidence supporting this claim is presented in Section 5.1.

## 2.2 Model-Based Reinforcement Learning

MBRL involves two steps. In the model learning step, the agent uses collected data to train a predictive model of the environment's dynamics. In the policy optimization step, the agent uses this model to simulate the environment to improve the policy. Although real experience could also be used for training, modern MBRL methods often rely solely on simulated trajectories for policy optimization. Our work also follows this pipeline and is closely connected with STORM (Zhang et al., 2023) and DreamerV3 (Hafner et al., 2023). Since this part of the literature review often repeats in recent MBRL works, we leave these details in the Appendix B for readers to reference.

## 3 Method

Figure 2 shows the full structure of our method. Our approach first employs self-supervised learning to model the environment's dynamics, then trains a model-free policy within the model's imagined trajectories. In this section, $\phi$, $\psi$, and $\theta$ denote the world model parameters, the critic (value) network parameters, and the actor (policy) network parameters, respectively. Additionally, $L$ refers to the batch length of the sampling or imagination trajectory segments, and $T$ is the length of an episode.

### 3.1 Object Feature and Visual Input

Our model leverages the first 8 output object memory features generated by Cutie's object transformer (Cheng et al., 2023), as described in Section 2.1. For visual input, we resize the original observation to a resolution of $64 \times 64$, following previous settings (Hafner et al., 2023; Zhang et al., 2023). The inputs are described by the following equations, where $t$ denotes the timestep, and $K$ represents the number of objects within the observation:

$$
\begin{aligned}
&\text{Observation:} && o_t \in \mathbb{R}^{3 \times H \times W}, \\
&\text{Object features:} && s_t^{\text{object}} = \text{Cutie}(o_t) \in \mathbb{R}^{K \times 2048}, \\
&\text{Visual input:} && s_t^{\text{visual}} = \text{Resize}(o_t) \in \mathbb{R}^{3 \times 64 \times 64}.
\end{aligned}
\tag{1}
$$

The value of $K$ is specific to the environment and predetermined by the user. For example, in Atari Pong, we set $K = 3$ to account for the two paddles and one ball. Additionally, though not explicitly stated in the equation, Cutie maintains internal states to retain information from previous observations, improving tracking consistency. These states are reset at the start of each episode.

### 3.2 Categorical VAE

Modelling an autoregressive sequence model on raw inputs often results in compounding errors (Hafner et al., 2023; Zhang et al., 2023; Alonso et al., 2024). To mitigate this, we employ a categorical VAE (Kingma & Welling, 2014; Hafner et al., 2023), which transforms input states $s_t$ into a discrete latent stochastic variable $z_t$, as formulated in Equation 2. The VAE encoder ($q_\phi$) and decoder ($p_\phi$) are implemented as multi-layer perceptrons (MLPs) for object feature vectors and convolutional neural networks (CNNs) for visual observations:

$$
\begin{aligned}
&\text{Encoder:} && z_t \sim q_\phi(z_t | s_t) \in \mathbb{R}^{K \times 16 \times 16} \text{ or } \mathbb{R}^{32 \times 32}, \\
&\text{Decoder:} && \hat{s}_t = p_\phi(z_t) \in \mathbb{R}^{K \times 2048} \text{ or } \mathbb{R}^{3 \times 64 \times 64}.
\end{aligned}
\tag{2}
$$

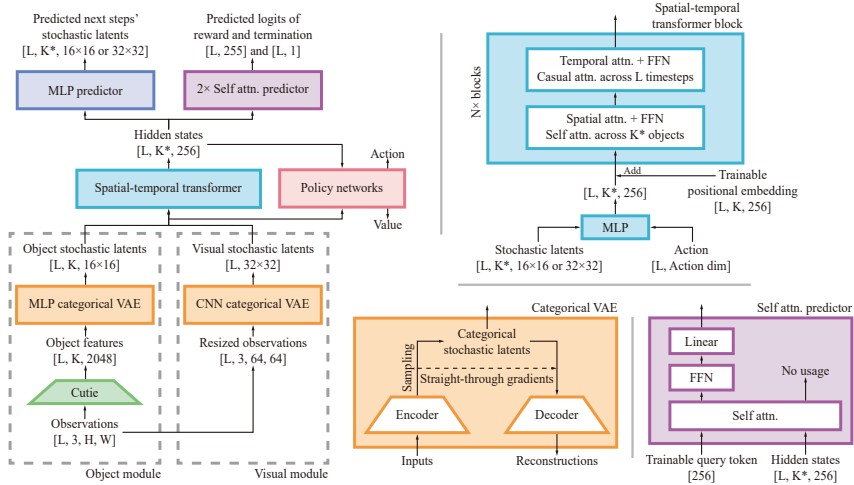

Figure 2: The model structure of our proposed OC-STORM. The tuples in square brackets represent the shapes of the corresponding tensors, where $L$ denotes the batch length or sequence length, $K$ is the number of objects, and $H$ and $W$ are the image height and width, respectively. The object module constitutes the proposed object-centric component, while the visual module processes resized raw observations. $K^*$ is explained in Section 3.3. The trainable token and positional embeddings are broadcasted to match the shapes of the corresponding tensors. The reward logit is 255-dimensional and used for the symlog two-hot loss (Hafner et al., 2023).

As sampling from a distribution lacks gradients for backpropagation, we apply the straight-through gradient trick (Bengio et al., 2013; Hafner et al., 2021) to retain them. The VAE treats each of the $K$ objects independently. Each latent variable comprises 16 categories with 16 classes for an object and 32 categories with 32 classes for the visual input. The configuration of $32 \times 32$ is inherited from prior work (Hafner et al., 2023; Zhang et al., 2023; Robine et al., 2023), while the $16 \times 16$ design is motivated by the fact that a single object contains less information than the entire scene.

## 3.3 Spatial-Temporal Transformer

The spatial-temporal transformer is designed to predict the future states of objects. Each transformer block contains a spatial attention block and a causal temporal attention block. Spatial attention among objects $(z_t^1, z_t^2, \ldots, z_t^K)$ facilitates understanding inter-object relationships within a timestep. Causal temporal attention across timesteps $(z_1^i, z_2^i, \ldots, z_T^i)$ predicts an object's future trajectory. We concatenate the actions with the object and visual states to provide the control signal. The spatial-temporal transformer is formulated as follows, where $h$ represents the hidden states or the transformer output, and $1 : L$ denotes timesteps from 1 to $L$:

$$\text{Transformer:} \quad h_{1:L} = f_\phi(z_{1:L}^{\text{object}}, z_{1:L}^{\text{visual}}, a_{1:L}),$$
$$h_{1:L} \in \mathbb{R}^{K^* \times L \times 256}. \tag{3}$$

The model can utilize either or both the object features and the visual input, with the visual input treated as an object during processing. We use $K^*$ to account for the variability in the number of objects due to different input choices. Specifically, $K^*$ can be $K$ (object module only), $K + 1$ (both modules), or $1$ (visual module only, which corresponds to the baseline STORM).

## 3.4 Prediction Heads

The hidden states generated by the transformer are used to predict environment dynamics, rewards, and termination signals. The dynamics predictor $g_\phi^{\text{Dyn}}$ is an MLP that predicts the distribution of the next step's latent variable. The reward and termination predictors $g_\phi^{\text{Reward}}$ and $g_\phi^{\text{Termination}}$ are

self-attention mechanisms, with structures depicted in Figure 2. A query token gathers information from multiple objects, similar to the `CLS` token in natural language processing (Devlin et al., 2019). The predictors are formulated as follows:

$$
\begin{aligned}
\text{Dynamics predictor:} \quad & \hat{z}_{t+1} \sim g_\phi^{\text{Dyn}}(\hat{z}_{t+1}|h_t), \\
\text{Reward predictor:} \quad & \hat{r}_t = g_\phi^{\text{Reward}}(h_t), \\
\text{Termination predictor:} \quad & \hat{\tau}_t = g_\phi^{\text{Termination}}(h_t).
\end{aligned}
\tag{4}
$$

### 3.5 Training of the World Model and the Policy

The world model is trained in a self-supervised manner, optimizing it end-to-end. The policy is trained over simulated trajectories generated by the world model and is optimized with a model-free actor-critic algorithm. Our setup closely follows DreamerV3 (Hafner et al., 2023), which is also similar to other MBRL methods (Zhang et al., 2023; Micheli et al., 2023; Robine et al., 2023). Full details are provided in Appendix D.

## 4 Experiments

We first evaluate the performance of our method on the Atari 100k benchmark (Bellemare et al., 2013), which serves as a standard testbed for measuring the sample efficiency of MBRL methods (Kaiser et al., 2020; Micheli et al., 2023; Hafner et al., 2023). We then further test our method on Hollow Knight (TeamCherry, 2017), which is a highly acclaimed game released in 2017. The core gameplay of Hollow Knight revolves around world exploration and combat with enemies, and we focus on combat with bosses in this work. Compared to Atari games, Hollow Knight's boss fights are visually more complex, with most key information representable as objects, making it well-suited to demonstrating the capabilities of our proposed pipeline.

As outlined in Section 3, our method can utilize either object features, visual observations, or both. In this section, all reported results from our method incorporate both types of inputs. A more detailed analysis of input selection will be presented in Section 5.2.

### 4.1 Atari 100k

We adhere to the Atari 100k settings established in previous work Bellemare et al. (2013); Hafner et al. (2023); Alonso et al. (2024); Zhang et al. (2023). In Atari, 100k samples correspond to approximately 1.85 hours of real-time gameplay. For each environment, we conduct five experiments using different random seeds. Each seed's performance is evaluated by the mean return across 20 episodes, and we report the average of these five mean episode returns. The human normalized score (HNS) is calculated with $(\text{score} - \text{random\_score})/(\text{human\_score} - \text{random\_score})$.

Table 1: Game scores and overall human-normalized scores on the selected games in the Atari 100k benchmark. The detailed results for each environment and the number of annotated objects are reported in Apppendix A.

| Game | Random | Human | IRIS | DreamerV3 | STORM | STORM* | OC-STORM |
|------|--------|-------|------|-----------|-------|--------|----------|
| HNS mean | 0% | 100% | 105% | 125% | 122% | 114.2% | **134.8%** |
| HNS median | 0% | 100% | 29% | **49%** | 42% | 42.5% | 43.8% |

The results are shown in Table 1. Overall, OC-STORM outperforms STORM on 18 of 26 tasks. To further assess the effectiveness of our method, we categorize the 26 games into two groups: games in which all relevant information can be captured by objects, and games in which some background information may be helpful. This categorization of environments is listed in Table 2. For environments where key elements can be primarily represented as objects, we find OC-STORM

significantly outperforms the baseline. For environments requiring a deeper understanding of background information, OC-STORM performs on par with the baseline. This confirms the value of our object-centric approach.

## 4.2 Hollow Knight

While the Atari benchmark is widely used in the reinforcement learning community, it has limitations for evaluating an object-centric approach. First, many Atari games require detailed background information, such as boundaries, terrains, and minimaps, which may not be easily represented as distinct objects. Second, some games have duplicate entities with identical appearances, which Cutie inherently struggles to differentiate. Lastly, Atari's visual simplicity allows methods like DreamerV3 and

Table 2: Human normalized mean of two categories in Atari 100k. OC-STORM outperforms the baseline in games that can be represented as objects and is on par with the baseline in other games.

| Category | Games | STORM* | OC-STORM |
|---|---|---|---|
| All key information for the decision can be represented as objects | Assault, Asterix, BankHeist, Breakout, Boxing, ChopperCommand, DemonAttack, Freeway, Jamesbond, Kangaroo, KungFuMaster, Pong, RoadRunner, Seaquest, UpNDown | 116.5% | **142.8%** |
| Not all key information for the decision can be represented as objects with Cutie | Alien, Amidar, BattleZone, CrazyClimber, Frostbite, Gopher, Hero, Krull, MsPacman, PrivateEye, Qbert | **130.0%** | 124.0% |

STORM to capture environments and backgrounds almost perfectly, but such simplicity would be rarely seen in real-world scenarios. In contrast, the boss fights in Hollow Knight offer a more suitable testbed, where the visuals are much more complex, including dynamic and distracting elements.

For Hollow Knight, we similarly limit the number of samples to 100k, equivalent to approximately 3.1 hours of real-time gameplay at 9 FPS. For each boss, we conduct 3 experiments with different random seeds. Each seed's performance is measured by the mean episode return across 20 runs, and the average of these 3 mean returns is reported.

Since Hollow Knight is not yet an established benchmark, existing methods differ significantly in sample step limits, resolution, environment wrapping, reward functions, boss selection, etc. This makes direct comparisons with existing methods challenging. As the primary goal of this work is to improve MBRL through the use of object-centric representations, we therefore compare our results with the equivalent baseline algorithm STORM. Nevertheless, we include the results from Yang (2023) on the boss Hornet Protector for a rough comparison in Appendix E.5.

Table 3: Episode returns and win rates (WR) of STORM and the proposed OC-STORM on Hollow Knight. The "#Objects" column shows the number of annotated objects for a boss. Scores that are the highest or within $5\%$ of the highest score are highlighted in bold. We provide training curves in Appendix E.6.

| Boss name | Random | Optimal | STORM | OC-STORM | STORM WR | OC-STORM WR | #Objects |
|---|---|---|---|---|---|---|---|
| God Tamer | 10 | 56 | 35.0 | **41.7** | **70.0%** | 55.0% | 4 |
| Hornet Protector | 7 | 37 | 28.1 | **32.4** | 66.7% | **100.0%** | 2 |
| Mage Lord | 3 | 38 | 19.6 | **28.0** | 5.0% | **48.0%** | 3 |
| Mantis Lords | 2 | 42 | 33.2 | **35.2** | 71.7% | **83.3%** | 3 |
| Mawlek | 8 | 41 | **36.9** | **37.2** | **98.3%** | **98.3%** | 3 |
| Pure Vessel | 2 | 55 | 8.9 | **15.7** | 0.0% | 0.0% | 2 |
| Pure Vessel (400k) | 2 | 55 | 25.3 | **35.0** | 6.7% | **13.3%** | 2 |

As seen in Table 3, though the original STORM can also learn a good policy on Hollow Kight, our proposed object-centric method converges significantly faster and yields stronger performance in most cases, especially when the environment is more challenging, such as for Mage Lord and Pure Vessel. Additionally, to evaluate the upper limit of our agent, we conduct a 400k run on Pure Vessel, which demonstrates that our agent can defeat one of the most difficult bosses in the game with enough training.

# 5 Analysis

## 5.1 Completeness of the Object Representation

As described in Section 2.1, we utilize the output feature of Cutie's object transformer. While this feature theoretically contains all the state and positional information of an object, it is uncertain whether it fully captures these details in practice. Specifically, we need to determine if the masked pooling could potentially obscure positional information. The agent's performance, as demonstrated in Section 4, provides general quantitative evidence. Here, we present qualitative evidence to support this claim.

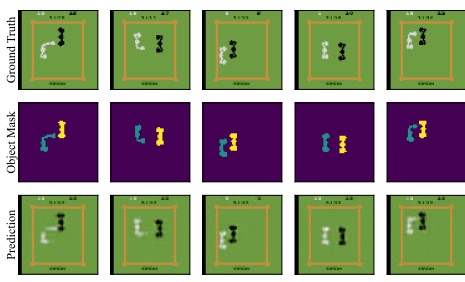

Figure 3: Observation reconstructions on Atari Boxing with two object feature vectors as inputs. The object mask row is generated using Cutie, which highlights the relevant objects.

To validate the completeness of the object representation, we trained a 4-layer ConvTranspose2d (Zeiler et al., 2010) decoder on the Atari Boxing game. This decoder takes two 2048-dimensional object features as inputs, corresponding to the white and black players, respectively, to reconstruct the observation. The dataset was collected using a random policy, with 10,000 frames for training and 1,000 frames for validation. Sample reconstructions result from the validation set are shown in Figure 3. This indicates that these features effectively capture the state and position of the objects.

## 5.2 Choice of the Object Representation

Cutie offers compact vector representations of objects, which are utilized in our method. Another option would have been to directly utilize the generated mask as part of our input, as in FOCUS (Ferraro et al., 2023). To assess the effectiveness of using feature vectors versus masks for object representation, we conduct an ablation study, with results displayed in Figure 4.

Figure 4 shows that the vector-based representation generally results in stronger performance than the mask-based representation. [1] In some environments, using only the vector representation leads to faster convergence than incorporating the visual module. Combining both modules offers consistent improvements across most environments, particularly in Hollow Knight.

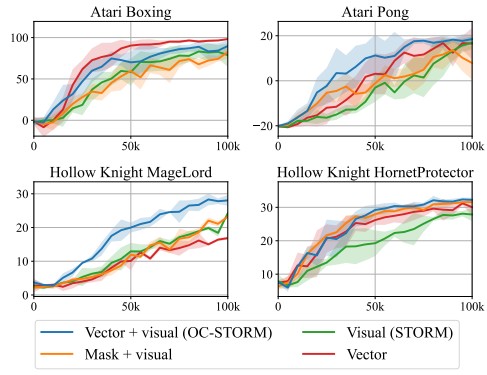

The main reason for using both vector and visual modules is that in some environments, the vector generated by Cutie may lose information without providing access to the visual observation as well. The mask representation may perform worse since downsampling the model-generated masks to STORM's $64 \times 64$ could make them excessively

Figure 4: Training episode returns for different input module configurations.

coarse, but using high-resolution visual input would significantly increase computational cost. In contrast, the vector representation is summarized from high-resolution input, which is more consistent, fine-grained and computationally efficient.

---

[1] We use a solid line to represent the mean of 5 seeds and use a semi-transparent background to represent the standard deviation. "Vector" and "visual" correspond to the object module and visual module, respectively, as depicted in Figure 2.

### 5.3 Analysis of Segmentation Model Errors

To mimic segmentation model failure, we randomly set the object feature vector to 0. This operation is identical to how we process the feature when Cuite detects nothing, as described in Appendix G.2. We conduct experiments on Atari Boxing and Pong with different zeroing probabilities. To avoid interference from visual inputs, we only use the object module in these experiments.

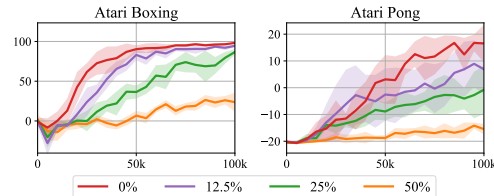

Figure 5: Training curves for Atari Boxing and Pong with 4 different zeroing probabilities.

The results are presented in Figure 5. As the detection accuracy of the vision model increases, the agent's performance improves accordingly. This also demonstrates the robustness of OC-STORM in handling unstable detection results. Additionally, since the zeroing process is purely random and the agent is trained only after the termination of each episode, every new episode during training serves as an indicator of test-time failure performance.

### 5.4 Additional Experiments and Analysis

To further demonstrate the effectiveness of OC-STORM, we also conduct experiments on Metaworld, a continuous control robotics benchmark, in Appendix F. The results demonstrate promising sample efficiency, providing evidence that OC-STORM can perform well on novel object-centric environments out-of-the-box, without significant adaptation of the pipeline or extensive tuning. We also include additional analysis on policy design and few-shot labeling, as well as an overview of the computational overhead of OC-STORM.

## 6 Conclusions and Limitations

In this work, we introduced OC-STORM, a MBRL pipeline designed to improve sample efficiency in visually complex environments. By integrating recent advances in object segmentation and detection, we mitigate the limitations of traditional reconstruction-based MBRL methods, which may be dominated by large background areas and overlook decision-relevant details. Through experiments on Atari and Hollow Knight, we demonstrated that object-centric learning could be successfully implemented without relying on internal game states or extensive labelling, highlighting the adaptability of our method to complex, visually rich environments. OC-STORM represents a meaningful step toward combining modern computer vision with reinforcement learning, offering an efficient framework for training agents in visually complex settings.

Our method has two main limitations, each of which corresponds to a potential future direction:

1. **Duplicated instances:** Current video object segmentation algorithms are primarily developed and trained to track a single object. When a scene contains two or more identical or similar objects, approaches like Cutie (Cheng et al., 2023) may fail to segment each object correctly and thus may affect performance.
2. **Background representation:** Our object representations do not capture elements that cannot be easily described as objects or compact vectors, such as walls, map boundaries, or the overall scene layout. Though we included visual inputs, the agent may still lose critical information for decision-making if those parts are small in pixel areas. This is a general limitation of any object-centric representation method.

These limitations are further illustrated and explained in Appendix I.

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

# Supplementary Materials

*The following content was not necessarily subject to peer review.*

## A    Full results on Atari 100k

Table 4: Game scores and overall human-normalized scores on the selected games in the Atari 100k benchmark. The "#Objects" column shows the number of annotated objects for an environment. Scores that are the highest or within $5\%$ of the highest score are highlighted in bold. STORM* denotes the results of re-running STORM using our codebase. Compared to the original version, we use a more lightweight configuration for faster training and decision-making on Hollow Knight. STORM* shares an identical configuration to the proposed OC-STORM, except for module usage. The higher score between OC-STORM and the baseline STORM* is underlined.

| Game | Random | Human | IRIS | DreamerV3 | STORM | DIAMOND | STORM* | OC-STORM | #Objects |
|---|---|---|---|---|---|---|---|---|---|
| Alien | 228 | 7128 | 420 | **1118** | 984 | 744 | 748.2 | **1101.4** | 4 |
| Amidar | 6 | 1720 | 143 | 97 | 205 | **226** | 144 | 162.7 | 2 |
| Assault | 222 | 742 | **1524** | 683 | 801 | **1526** | 1376.7 | 1270.4 | 4 |
| Asterix | 210 | 8503 | 854 | 1062 | 1028 | **3698** | 1318.5 | 1753.5 | 3 |
| BankHeist | 14 | 753 | 53 | 398 | 641 | 20 | 990 | **1075.2** | 3 |
| BattleZone | 2360 | 37188 | 13074 | **20300** | 13540 | 4702 | 5830 | 4590 | 3 |
| Boxing | 0 | 12 | 70 | 82 | 80 | 87 | 81.2 | **92.2** | 2 |
| Breakout | 2 | 30 | 84 | 10 | 16 | **132** | 41 | 52.5 | 3 |
| ChopperCommand | 811 | 7388 | 1565 | **2222** | 1888 | 1370 | 1644 | 2090 | 4 |
| CrazyClimber | 10780 | 35829 | 59234 | 86225 | 66776 | **99168** | 79196 | 84111 | 2 |
| Demon Attack | 152 | 1971 | **2034** | 577 | 165 | 288 | 324.6 | 411.3 | 4 |
| Freeway | 0 | 30 | 31 | 0 | 0 | **33** | 0 | 0 | 2 |
| Frostbite | 65 | 4335 | 259 | **3377** | 1316 | 274 | 365.9 | 259.6 | 3 |
| Gopher | 258 | 2413 | 2236 | 2160 | **8240** | 5898 | 5307.2 | 4456.8 | 2 |
| Hero | 1027 | 30826 | 7037 | **13354** | 11044 | 5622 | 11434.1 | 6441.4 | 2 |
| James Bond | 29 | 303 | 463 | **540** | 509 | 427 | 408 | 347 | 4 |
| Kangaroo | 52 | 3035 | 838 | 2643 | 4208 | **5382** | 3512 | 4218 | 4 |
| Krull | 1598 | 2666 | 6616 | 8171 | 8413 | 8610 | 6522.2 | **9714.6** | 2 |
| KungFuMaster | 256 | 22736 | 21760 | **25900** | 26182 | 18714 | 20046 | 24988 | 3 |
| MsPacman | 307 | 6952 | 999 | 1521 | **2673** | 1958 | 1489.5 | 2400.7 | 2 |
| Pong | -21 | 15 | 15 | -4 | 11 | **20** | 18.4 | 20.6 | 3 |
| PrivateEye | 25 | 69571 | 100 | 3238 | **7781** | 114 | 100 | 85 | 3 |
| Qbert | 164 | 13455 | 746 | 2921 | **4522** | **4499** | 2910.5 | 4546.2 | 3 |
| RoadRunner | 12 | 7845 | 9615 | 19230 | 17564 | **20673** | 14841 | 20482 | 4 |
| Seaquest | 68 | 42055 | 661 | **962** | 525 | 551 | 557.4 | 712.2 | 3 |
| UpNDown | 533 | 11693 | 3546 | **46910** | 7985 | 3856 | 6127.9 | 6623.2 | 3 |
| HNS mean | 0% | 100% | 105% | 125% | 122% | **146%** | 114.2% | 134.8% | |
| HNS median | 0% | 100% | 29% | **49%** | 42% | 37% | 42.5% | 43.8% | |

## B    Related Work

### B.1    Model-based reinforcement learning

Ha & Schmidhuber (2018) first demonstrated the feasibility of learning by imagination in pixel-based environments. SimPLe (Kaiser et al., 2020) further extended this idea to Atari games (Bellemare et al., 2013), though with limited efficiency. The Dreamer series (Hafner et al., 2019; 2021; 2023) employs categorical variational autoencoders and recurrent neural networks (RNNs), to achieve robust performance across diverse domains. Dreamer introduces both a stable discretization method and a set of techniques for robust optimization across domains with diverse observations, dynamics, rewards, and goals. TWM (Robine et al., 2023) and STORM (Zhang et al., 2023) replace the RNN sequence model in Dreamer with transformers, enhancing parallelism during training. TWM encodes the observation, reward, and termination as three input tokens for the transformer, while STORM encodes them as a single token, demonstrating better efficiency. IRIS (Micheli et al., 2023), and improved efficiency variants Δ-IRIS (Micheli et al., 2024) and REM (Cohen et al., 2024), utilize

VQ-VAE (van den Oord et al., 2017) for multi-token latent representations. DIAMOND (Alonso et al., 2024) employs a diffusion process as the world model, further improving the final performance. All of these methods predominantly use an $L_2$ reconstruction loss for self-supervised learning.

## B.2 Object-Centric Reinforcement Learning

Object-centric learning has gained increasing attention in both the machine learning and cognitive psychology fields (Driess et al., 2023; Delfosse et al., 2023). Human infants inherently possess an understanding of objects (Spelke, 1990), suggesting that object extraction from visual observations may be fundamental for high-level decision-making. From a data processing perspective, utilizing object information can significantly reduce computational costs compared to raw visual inputs.

Many attempts have been made to introduce object-centric learning to reinforcement learning systems. However, to our knowledge, no existing methods could be directly applied to Atari games or Hollow Knight without leveraging internal game states or an extensive number of annotations. These object-centric learning methods broadly follow two main trends: two-stage and end-to-end.

**Two-Stage** methods usually first use computer vision models or techniques to detect objects, then train the policy based on this object-level information. Current approaches often require labour-heavy task-specific fine-tuning (Devin et al., 2018; Liu et al., 2021), access to game memories (Delfosse et al., 2023; Jain, 2024), or leverage game-specific observation structures (Stanic et al., 2024). FOCUS (Ferraro et al., 2023), the most similar work to ours, is a model-based method that uses TrackingAnything (Yang et al., 2023) to generate segmentation masks, which are then fed into DreamerV2 (Hafner et al., 2021) for policy training. However, using binary masks for object representation limits efficiency, which will be discussed in Section 5.2. Moreover, FOCUS has only been tested on six robot control tasks and hasn't been fully explored in more visually complex environments.

**End-to-End** methods jointly learn object perception and policy, often using unsupervised slot-based approaches (Locatello et al., 2020) to discover and represent objects. While these methods allow the visual module to be trained alongside the world model or policy network, their unsupervised learning nature leads to poor object detection quality, especially in noisy, real-world scenes. As a result, they are typically limited to simple object-centric benchmarks (Watters et al., 2024; Ahmed et al., 2021) and struggle to generalize to visually complex tasks. Several model-based (Veerapaneni et al., 2019; Lin et al., 2020; van Bergen & Lanillos, 2022) and model-free (Yoon et al., 2023; Haramati et al., 2024) algorithms have used these ideas. Nakano et al. (2024) added slot attention to STORM, achieving stronger performance on the OCRL benchmark (Yoon et al., 2023). Our work also builds on STORM, but we use a pre-trained vision model instead of unsupervised slot attention, allowing us to better handle more visually complex environments.

## B.3  Review of Object Representation Methods

Table 5: A brief review of state-of-the-art methods in different fields of computer vision related to object-centric reinforcement learning.

| Method | Description |
| --- | --- |
| Cutie | Retrieval-based semi-supervised video object segmentation algorithm. Provides compact vector representation of objects using an object transformer (Cheng et al., 2023). |
| XMem | Retrieval-based semi-supervised video object segmentation algorithm with multi-level memory system. Lacks compact object representation (Cheng & Schwing, 2022). |
| SAM | Open-set image segmentation algorithm. Generates masks via user prompts but requires a prompt for each frame in a video (Kirillov et al., 2023). |
| TrackAnything | SAM + XMem for fine-grained video segmentation. Requires double amount of computing compared to XMem (Yang et al., 2023). |
| PerSAM | One-shot enhancement of SAM. Difficult to expand to few-shot cases (Zhang et al., 2024). |
| SAM2 | Semi-supervised video object segmentation algorithm. Capable of providing compact vector representations. Potentially being more efficient than Cutie on object-centric MBRL (Ravi et al., 2024). |
| YOLO | Closed-set object detection algorithm. Requires extensive annotations for training or fine-tuning (Redmon et al., 2016; Jocher et al., 2023). |
| GroundingDINO | Open-set object detection algorithm. Generates object bounding boxes using natural language prompts but struggles with rare or abstract cases, such as video game players (Liu et al., 2023). |
| Slot attention | Unsupervised image object discovery and segmentation algorithm. Underperforms compared to supervised methods (Locatello et al., 2020). |
| SAVi | Unsupervised video object discovery and segmentation algorithm. Evaluation is semi-supervised, though training is unsupervised. Underperforms compared to semi-supervised methods (Kipf et al., 2022; Elsayed et al., 2022). |
| Omnimotion | Video point-tracking algorithm. Provides pixel-level tracking but lacks object-level information and does not support few-shot scenarios (Wang et al., 2023). |
| Unimatch | Dense optical flow and depth estimation algorithm. Useful for moving identity detection but cannot extract object-level information and struggles with out-of-domain generalization (Xu et al., 2023). |

SAM2 (Ravi et al., 2024) provides a strong alternative to Cutie, and could also be integrated into our proposed pipeline. As it was published concurrently with this work, we leave the investigation to future work.

## C    Reconstruction Analysis in STORM

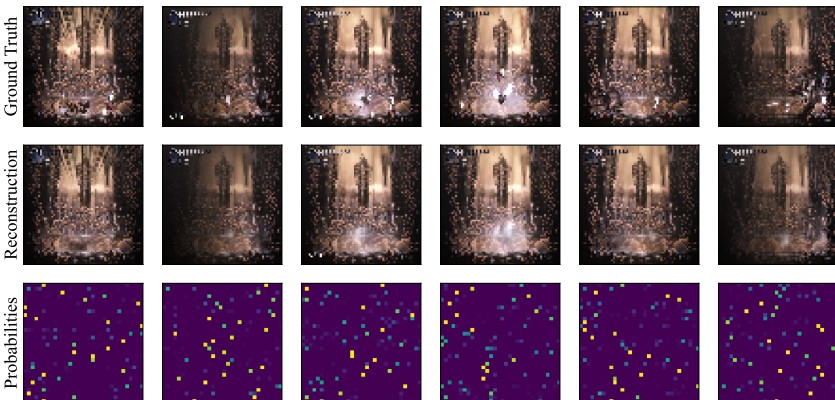

Figure 6: Sample ground truth observations from the Hollow Knight boss Hornet Protector, the reconstruction results of STORM, and the probabilities of the $32 \times 32$ latent distribution. In this instance, STORM was trained on 200k samples. The key characters are missing in the reconstructions.

As mentioned in Section 1, MBRL methods that rely on $L_2$ reconstruction loss may miss key elements for controlling. Here, we present a qualitative reconstruction example using STORM (Zhang et al., 2023), as shown in Figure 6. Since the $64 \times 64$ image processed by the model may be hard to interpret, a high-resolution sample is provided in Figure 7.

The two main characters in the game are the Knight on the left (in white and black) and the boss Hornet Protector on the right (in red). The 9 "masks" in the top left represent the Knight's remaining health, showing 1 health point remaining and 8 lost in the case depicted in Figure 7.

The autoencoder captures static or large-area features, such as lighting, shadows, streaks, smoke, and health indicators, which are not crucial for gameplay or rewards. However, the model struggles with character positions and states. While MBRL methods have shown nearly perfect reconstruction and simulation in some simpler environments like Atari games (Micheli et al., 2023; Zhang et al., 2023), they face challenges in visually complex environments such as Hollow Knight or real-life scenes. Similar difficulties could also be observed in Minecraft (Guss et al., 2021), as depicted in the DreamerV3 paper (Hafner et al., 2023).

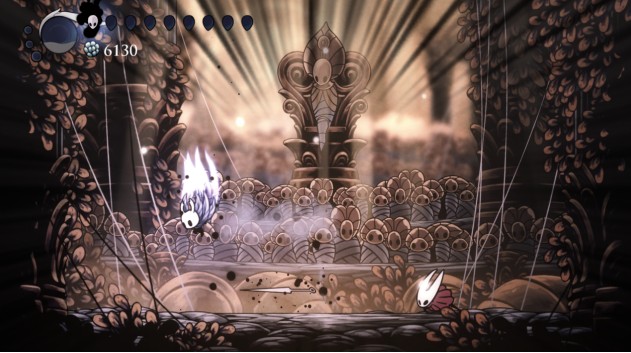

Figure 7: Sample high-resolution frame from the Hollow Knight boss Hornet Protector. Though not visible in this figure, the background is dynamic, which adds to the challenge of learning for the world model.

Despite missing key objects in reconstructions, MBRL algorithms (Zhang et al., 2023; Hafner et al., 2023) that learn solely from generated trajectories still achieve reasonable control in these tasks. The

precise reasons for this remain unclear. One possible explanation is that the encoder, even without perfect reconstructions, can differentiate character states and generate distinct latent distributions, as illustrated in Figure 6.

These results indicate that even with reward and termination supervision, the world model struggles to prioritize key objects. Simply increasing resolution may not help, as the reconstruction loss still weighs characters proportionally to the whole scene, potentially inflating computation and memory costs. Increasing latent variables, as in IRIS (Micheli et al., 2023) or using multi-step diffusion, as in DIAMOND (Alonso et al., 2024), could improve performance but is computationally expensive.

Thus, our proposed object-centric representation offers an effective solution to these challenges.

## D  Loss Functions

### D.1  World Model Learning

The world model is trained in a self-supervised manner, optimizing it end-to-end. Our setup closely follows DreamerV3 (Hafner et al., 2023), with details presented for completeness. We use mean squared error (MSE) loss for reconstructing the original inputs, symlog two-hot loss $\mathcal{L}_{\text{sym}}$ (Hafner et al., 2023) for reward prediction, and binary cross-entropy (BCE) loss for termination signal prediction. These losses, collectively referred to as prediction loss, are defined as:

$$\mathcal{L}_{\text{pred}}(\phi) = \underbrace{||\hat{s}_t - s_t||_2^2}_{\text{Reconstruction Loss}} + \underbrace{\mathcal{L}_{\text{sym}}(\hat{r}_t, r_t)}_{\text{Reward Loss}} + \underbrace{\tau_t \log \hat{\tau}_t + (1 - \tau_t) \log(1 - \hat{\tau}_t)}_{\text{Termination Loss}}. \tag{5}$$

The dynamics loss $\mathcal{L}_t^{\text{dyn}}(\phi)$ guides the sequence model in predicting the next distribution. The representation loss $\mathcal{L}_t^{\text{rep}}(\phi)$ allows the encoder's output to be weakly influenced by the sequence model's prediction, ensuring that the dynamics are not overly difficult to learn. These losses are identical Kullback–Leibler (KL) divergence losses except for their gradient propagation settings. We use $\text{sg}(\cdot)$ to denote the stop gradient operation. The dynamics and representation losses are defined as:

$$\mathcal{L}_{\text{dyn}}(\phi) = \max\big(1, \text{KL}\big[\text{sg}(q_\phi(z_{t+1}|s_{t+1})) \,||\, g_\phi^{\text{Dyn}}(\hat{z}_{t+1}|h_t)\big]\big), \tag{6a}$$

$$\mathcal{L}_{\text{rep}}(\phi) = \max\big(1, \text{KL}\big[q_\phi(z_{t+1}|s_{t+1}) \,||\, \text{sg}(g_\phi^{\text{Dyn}}(\hat{z}_{t+1}|h_t))\big]\big). \tag{6b}$$

The $\max$ operation represents free bits for KL divergence, encouraging the model to focus on optimizing prediction losses for better feature extraction if the KL divergence is too small.

The total loss function for training the world model is calculated as follows, where $\mathbb{E}_{\mathcal{D}}$ denotes the expectation over samples from the replay buffer:

$$\mathcal{L}(\phi) = \mathbb{E}_{\mathcal{D}}\Big[\mathcal{L}_{\text{pred}}(\phi) + \mathcal{L}_{\text{dyn}}(\phi) + 0.5\mathcal{L}_{\text{rep}}(\phi)\Big]. \tag{7}$$

The coefficient of 0.5 for $\mathcal{L}_{\text{rep}}$ is used to prevent posterior collapse (Lucas et al., 2019), a situation where the model produces the same distribution for different inputs, causing the dynamics loss to trivially converge to 0. The imbalanced KL divergence loss helps to mitigate this issue.

### D.2  Policy Learning

The policy learning approach closely follows that of DreamerV3 (Hafner et al., 2023), with modifications specific to our method. The key differences lie in the input for the policy and the action dimension for the game Hollow Knight. We use the concatenation of object latents, object hidden states, visual latents, and visual hidden states as input features. For Hollow Knight, a multi-discrete action space is employed.

The agent learns entirely on the imagination trajectories generated by the world model. To begin the imagination process, we first sample a short contextual trajectory from the replay buffer. During imagination, future environmental inputs $s_{t+1:L}$ are unknown, and sampling from the posterior distribution $q_\phi(z_t|s_t)$ is unavailable. Thus, we sample the latent variable from the prior distribution $g_\phi^{\text{Dyn}}(\hat{z}_{t+1}|h_t)$ and optimize the policy over $\hat{z}_{t+1}$. However, during testing, the agent interacts directly with the environment, allowing access to the posterior distribution of the last observation. This introduces a difference in notation. For simplicity, we do not distinguish between $z_t$ and $\hat{z}_t$ in the following descriptions.

The agent uses both the latent variable $\hat{z}_t$ and hidden states $h_t$ as inputs, as defined below:

$$\text{Critic:} \quad V_\psi(z_t, h_t) \approx \mathbb{E}_{\pi_\theta, \phi}\Big[\sum_{k=0}^{T} \gamma^k r_{t+k}\Big], \tag{8}$$

$$\text{Actor:} \quad a_t \sim \pi_\theta(a_t|z_t, h_t).$$

Here, $T$ is the number of timesteps in the episode. We use two separate MLPs for the critic and actor networks. The symbol $\phi$ indicates that the trajectories are generated within the imagination process of the world model.

For value loss, we employ the $\lambda$-return $G_t^\lambda$ (Sutton & Barto, 2018; Hafner et al., 2023) to improve value estimation. It is recursively defined as follows, where $\hat{r}_t$ is the reward predicted by the world model, and $\hat{\tau}_t$ represents the predicted termination signal:

$$G_t^\lambda \doteq \hat{r}_t + \gamma(1 - \hat{\tau}_t)\Big[(1 - \lambda)V_\psi(z_{t+1}, h_{t+1}) + \lambda G_{t+1}^\lambda\Big], \tag{9a}$$

$$G_L^\lambda \doteq V_\psi(z_L, h_L). \tag{9b}$$

To regularize the value function, we maintain an exponential moving average (EMA) of the critic's parameters, as defined in Equation (10). This regularization technique stabilizes training and helps prevent overfitting, where $\psi_t$ represents the current critic parameters, $\sigma$ is the decay rate, and $\psi_{t+1}^{\text{EMA}}$ denotes the updated critic parameters:

$$\psi_{t+1}^{\text{EMA}} = \sigma\psi_t^{\text{EMA}} + (1 - \sigma)\psi_t. \tag{10}$$

For policy gradient loss, we apply return-based normalization for the advantage value. The normalization ratio $S$ is defined in Equation (11) as the range between the 95th and 5th percentiles of the $\lambda$-return $G_t^\lambda$ across the batch (Hafner et al., 2023):

$$S = \text{percentile}(G_t^\lambda, 95) - \text{percentile}(G_t^\lambda, 5). \tag{11}$$

The complete loss functions for the actor-critic algorithm are given by Equation (12):

$$\mathcal{L}(\theta) = \mathbb{E}_{\pi_\theta, \phi}\left[-\text{sg}\left(\frac{G_t^\lambda - V_\psi(s_t)}{\max(1, S)}\right)\ln \pi_\theta(a_t|z_t, h_t) - \eta H\big(\pi_\theta(a_t|z_t, h_t)\big)\right], \tag{12a}$$

$$\mathcal{L}(\psi) = \mathbb{E}_{\pi_\theta, \phi}\left[\mathcal{L}_{\text{sym}}\Big(V_\psi(z_t, h_t), \text{sg}(G_t^\lambda)\Big) + \mathcal{L}_{\text{sym}}\Big(V_\psi(z_t, h_t), \text{sg}\big(V_{\psi^{\text{EMA}}}(z_t, h_t)\big)\Big)\right]. \tag{12b}$$

Here, $H(\cdot)$ denotes the entropy of the policy distribution, and $\eta = 1 \times 10^{-3}$ is the coefficient for entropy loss.

## E   Hollow Knight

### E.1   Related Work

Despite its popularity among players, Hollow Knight has seen limited use as a benchmark for research in reinforcement learning. We introduce repositories, published research, and other relevant resources

that leverage or explore Hollow Knight as a benchmark. Cui (2021) employs DQN (Mnih et al., 2015) and its variants but requires modding the game background to black to enhance character perception. Yang (2023) uses the Rainbow algorithm (Hessel et al., 2018) with additional techniques like DrQ (Yarats et al., 2022; 2021), achieving high win rates against several of the game's bosses. Yang's repository has been widely forked and adopted. Building on his work, Lee (2023) studies the effect of reward shaping, while Sun (2024) focuses on improving training efficiency by tuning the game interaction configuration and switching to the PPO algorithm (Schulman et al., 2017). Jain (2024) leverages internal game states to extract hitboxes as input for the algorithm, representing them as segmentation masks that are passed to DQN or PPO.

### E.2 Environment Configuration

Hollow Knight is a modern video game developed with Unity (Technologies, 2005). To our knowledge, efficient simulators for this game, such as those available for Atari (Brockman et al., 2016; Towers et al., 2023), do not exist. Therefore, we developed a custom wrapper that captures screenshots of the game at 9 FPS and sends keyboard signals to execute actions. The 9 FPS rate is a choice based on the author's experience with the game and considerations for computational efficiency. To obtain reward signals, we developed a modding plugin (Bham & Wyza, 2017) that logs when the player-controlled character (the Knight) either hits an enemy or is hit. Our wrapper then parses this log file to generate reward and termination signals.

The game execution and agent training are conducted on a Windows machine. To monitor training progress and statistics without interrupting the game, we needed a method to send keyboard inputs to a background or unfocused window. However, Windows lacks an API for this purpose. As a result, the game window must remain in the foreground, fully occupying the training device and hindering monitoring. To address this, we utilized a Hyper-V (Cooley, 2022) Windows virtual machine to run the game in the background, with Ray (Moritz et al., 2018) facilitating communication between the host and virtual machine. Training and processing occur on the host machine, while the virtual machine handles interactions with the environment. This setup can be extended to distributed nodes, with some handling game rendering and others managing training tasks.

For in-game configuration, the charms (Wiki, 2018) are set to Unbreakable Strength, Quick Slash, Soul Catcher, and Shaman Stone across all experiments. This configuration is chosen to explore the agent's fighting potential rather than its glitch-finding abilities.

### E.3 Action Space

All previous works utilize a human-specified action space rather than the original keyboard inputs. For example, in the Yang (2023) implementation, short and long jumps are treated as two distinct actions, which are originally controlled by the duration of the jump button press. His environment wrapper handles this difference with a fixed command. While this design reduces the exploration and computational costs for reinforcement learning agents, it cannot capture the full range of possible actions in the game. Advanced operations, as demonstrated in this video (CrankyTemplar, 2018), require full control of the keyboard. Therefore, we design the action space as a multi-binary-discrete one that directly binds to the press and release of the physical keyboard, which will be explained in Section E.3.

We design the action space as a multi-binary-discrete space directly tied to the press and release states of eight specific keys on the keyboard. These keys include `W`, `A`, `S`, and `D` for movement directions, and `J`, `K`, `L`, and `I` for attack, jump, dash, and spell actions, respectively. Each key's state is represented as a binary variable, where 0 corresponds to a key release and 1 corresponds to a key press. The action can therefore be described as $a \in \{0, 1\}^8$, with each element representing the binary state of one key. The key's state is maintained between frames, and a toggle signal is sent only when there is a change in the key state from $0 \to 1$ (press) or $1 \to 0$ (release).

The probability of an action is determined by the independent probabilities of each key's state:

$$\pi_\theta(a|z,h) = \prod_{k=1}^{8} \pi_\theta(a^k|z,h) \tag{13}$$

where $a^k$ denotes the state of the $k$-th key.

The entropy of the action space, $H(\pi_\theta(a|z,h))$, is the sum of the entropies of the individual key states:

$$H(\pi_\theta(a|z,h)) = \sum_{k=1}^{8} H(\pi_\theta(a^k|z,h)) \tag{14}$$

This design provides fine-grained control over the agent's actions, allowing for the execution of complex manoeuvres while maintaining a tractable exploration space for reinforcement learning.

### E.4  Reward Shaping in Hollow Knight

Most existing methods (Yang, 2023; Sun, 2024; Jain, 2024; Cui, 2021; Lee, 2023) for Hollow Knight use a reward structure of +1 for hitting an enemy and -1 for taking damage. Some approaches modify the weighting ratios, while others introduce auxiliary rewards for performing specific actions. However, we found that these settings are suboptimal for training reinforcement learning agents.

Our method assigns a +1 reward signal for hitting an enemy and a virtual termination signal upon being hit. The game continues until the episode naturally ends. The termination signal is stored in the replay buffer for training the world model, treating health loss as a life-loss event. Leveraging life-loss information is a common technique that aids in value estimation (Ye et al., 2021; Micheli et al., 2023; Zhang et al., 2023; Alonso et al., 2024). Additionally, the Knight can damage enemies in multiple ways, and these damages are normalized against the base attack damage to compute the positive reward.

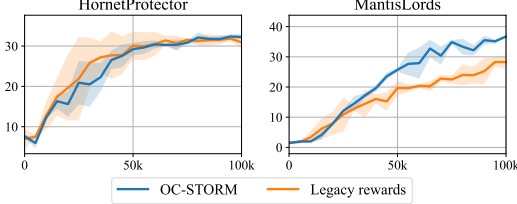

Figure 8: Training episode returns for Hollow Knight's Hornet Protector and Mantis Lords under different reward settings. "Legacy rewards" refer to the reward scheme used in prior works. For comparison, we aligned the returns from "legacy rewards" with our baseline settings by accounting for lost health.

Here, we present the key differences between the two reward settings. As illustrated in Figure 8, our reward configuration is more robust than those used in previous studies, resulting in significantly improved performance, especially in more challenging environments like Mantis Lords. This improvement can be analyzed from two perspectives:

1. Terminating the episode upon being hit better aligns with human cognition and the agent's expected behaviour. The aim is for the agent to deal as much damage as possible without taking any. While this may seem aggressive, raising concerns that the agent might sacrifice itself to deal more damage, neither our qualitative nor quantitative results show this tendency. Survival naturally offers more opportunities to deal future damage, which the agent learns to prioritize. Although applying a negative penalty for being hit could prevent the agent from sustaining multiple consecutive hits in highly unfavourable situations, such scenarios should not occur under an optimal or near-optimal policy.

2. While maintaining the same optimal policy, truncating future rewards upon being hit significantly reduces the variance in value estimation. Hollow Knight is a highly stochastic environment where bosses behave aggressively yet unpredictably. Estimating value directly over an episode (lasting approximately 300 to 700 timesteps) is inherently challenging in such settings.

### E.5   Comparision with a Model-Free Baseline

As introduced in Section E.1, Yang's repository (Yang, 2023) is a widely recognized implementation within the community. In this section, we compare the performance against the boss Hornet Protector.

Yang's reward structure assigns +0.8 for hitting an enemy and -0.8 for taking damage, with additional auxiliary rewards on the order of $1 \times 10^{-4}$ for various actions. A small feedback reward is also given at the end of each episode. The choice of a 0.8 weight factor for rewards reflects the use of +1/-1 reward clipping, with a margin reserved for the auxiliary rewards. We provide a broad comparison with this approach below.

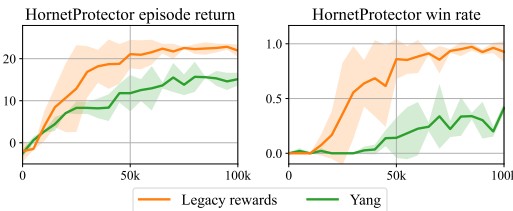

Figure 9: Training episode returns and win rates on Hollow Knight's Hornet Protector with our proposed method and Yang's (Yang, 2023) method. "Legacy rewards" are as described in Section E.4. We applied some preprocessing to align the two returns for easier comparison, so the "legacy rewards" curve may appear different from the one shown in the previous section. The win rate is more straightforward and can be used for comparison without changing.

As shown in Figure 9, our implementation is more efficient than Yang's. As we noted in Section 4.2, there are significant differences between our methods, making this **not necessarily** a fair comparison from an algorithmic standpoint. This comparison is intended solely to demonstrate the efficiency of our implementation.

Additionally, Yang claims that his agent can achieve 10 wins out of 10 battles, which is accurate despite his win rate in our plot appearing to be lower than 100%. Two reasons may lead to this. First, his original sample steps are greater than ours, which may account for differences in performance. Second, our in-game charm configuration (Wiki, 2018) differs from the one used in his implementation. When testing Yang's implementation, we retained our current charm settings, which likely impacted the win rate results.

## E.6 Training Curves

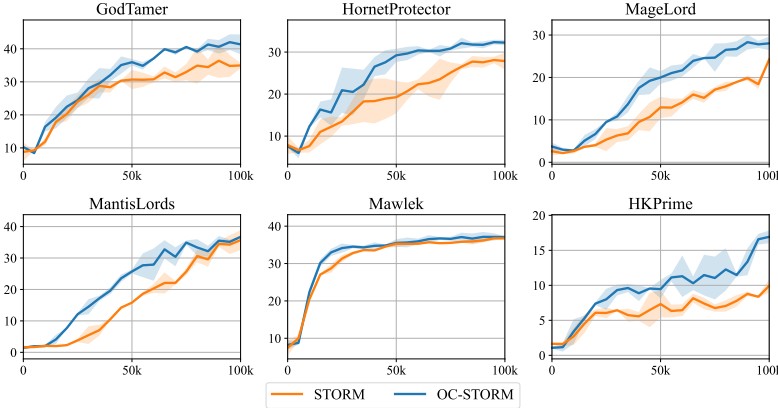

Figure 10: The training episode returns on Hollow Knight. We use a solid line to represent the mean of 3 seeds and use a semi-transparent background to represent the standard deviation.

## F   Additional Experiments and Analysis

### F.1   Additional Experiments on Meta-World

To evaluate the potential of OC-STORM on continuous control tasks, we conduct 4 experiments on the Meta-world benchmark. We compare our results with MWM (Seo et al., 2022), which is also designed to help the world model to focus on small dynamic objects. We choose 1 easy, 2 medium, and 1 hard task according to the MWM paper (see Seo et al. (2022) Appendix F, Experiments Details). These tasks are randomly selected to cover different objects and policies.

As shown in Figure 11, OC-STORM demonstrates improved sample efficiency on 3 of 4 tasks, providing evidence that this approach can also perform well on continuous tasks out-of-the-box, without significant adaptation of the pipeline or extensive tuning for these very different continuous control environments.

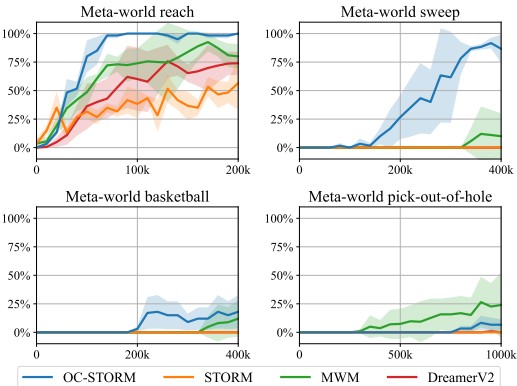

Figure 11: Training success rates on 4 Meta-world (Yu et al., 2019) tasks. The data of MWM (Seo et al., 2022) and DreamerV2 (Hafner et al., 2021) is from the MWM paper. OC-STORM generally exhibits higher sample efficiency than STORM. In some tasks, it also outperforms MWM in terms of efficiency and performance.

We provide the sample annotation masks used in our Meta-world exeriments below in Figure 12.

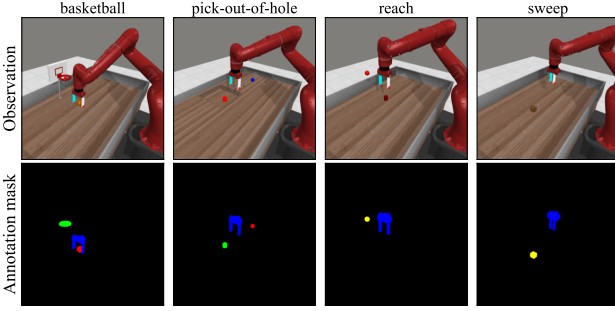

Figure 12: Sample frames and annotation masks for Meta-world tasks.

### F.2   Attention-Based policy or MLP-based Policy

When handling multiple objects, naturally one would think of an attention-based policy network like the self-attention predictor described in Figure 2. A previous work OC-SA (Stanic et al., 2024) has also explored such structure. However, we still design our actor and critic networks as MLPs which take the concatenation of object latent variables and hidden states as input.

We found that the attention-based policy tends to overfit pre-learned behaviours and makes it hard to learn new knowledge. This won't be a major issue in stationary games like Boxing but will face trouble in non-stationary games like Pong. For example, the attention-based policy can quickly learn how to catch the ball but can't efficiently learn how to score against the opponent. On the one hand, we can confirm that by visually checking the rendered episodes. On the other hand, numerically speaking, we can observe the episode length of playing Pong. If the episode length increases while the episode returns remain at the same level, then we can tell that the agent learns how to catch the ball, but is stuck in that local optimum.

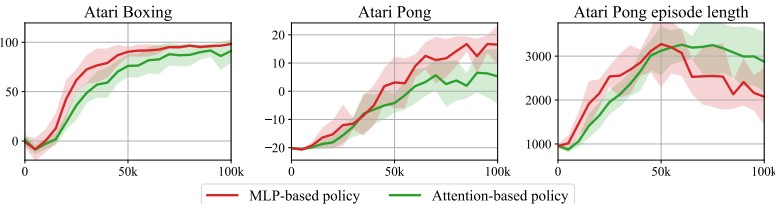

Figure 13: Training episode returns for Atari Boxing, Pong and episode lengths for Pong of attention-based policy and MLP-based policy. The attention-based policy can learn as quickly as the MLP-based policy for catching the ball but struggles to transition to the scoring phase in Atari Pong.

As the results plotted in Figure 13, we can tell that attention-based policy suffers from that issue. The episode length of both policies rises at a similar speed before 50k steps, but it declines slower for the attention-based policy after that. The experiments are conducted using only the object module, so the MLP-based policy curves are identical to the "vector" ones in Figure 4. As the visual latent itself contains all the information, the agent can choose only to use that part of the information and thus may affect our judgement on the effectiveness of the attention-based policy.

Though the attention-based policy has the potential to handle a dynamic number of objects, our experiments are conducted on a fixed number. As it doesn't demonstrate superior performance than the MLP-based policy in our case, we always use MLP-based in other tasks for consistency in evaluation.

### F.3 Impact of the Number of Annotations

Since Cutie is a retrieval-based algorithm that stores past frames and masks in a buffer for reference, it naturally supports the use of multiple annotation masks beyond the first frame by substituting model-generated masks in the buffer. Incorporating more label masks can capture a wider range of object states, leading to more consistent segmentation results. However, reducing the number of labels can further lower annotation costs and computational complexity. In this section, we explore the impact of the number of labels on agent performance. To eliminate the influence of visual input, we conduct these experiments using only the object module.

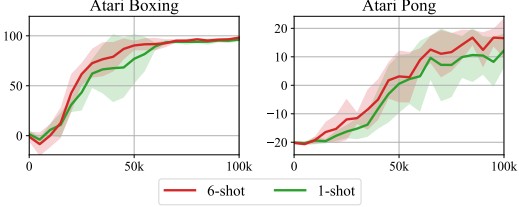

Figure 14: Training episode returns for Atari Boxing and Pong with different numbers of annotation segmentation masks. Increasing the number of annotation masks enhances the robustness of the agent's performance.

As shown in Figure 14, increasing the number of annotation segmentation masks enhances the robustness of the agent's performance, even in visually static environments like Atari Boxing and Pong. In these environments, a single frame can include all necessary objects for decision-making. However, Cutie may lose track of objects if their states deviate significantly from those in the labelled masks, such as the punching state versus the standing state in Boxing, or the paddle in Pong when it is partially off-screen and appears shorter than when centred.

Moreover, in complex environments like Hollow Knight and Minecraft, a single frame may not capture all objects, which often necessitates additional segmentation masks. For consistency in evaluation, we use six annotation segmentation masks for Atari and twelve for Hollow Knight.

### F.4 The Computational Overhead of OC-STORM

The computational overhead of OC-STORM on Atari games with an NVIDIA GeForce RTX 3090 is shown in Table 6. The input resolution for Cutie is 320*420 (double the original 160*210). Thus the computational cost of introducing Cutie is acceptable in many cases.

Table 6: The computational overhead of OC-STORM on Atari games. The three numbers in a block here mean sample or evaluation speed (iterations/second), training speed (iterations/second) and hours spent for a train with a 100k sample budget, respectively.

| Algorithm | 0 objects | | | 1 object | | | 2 objects | | | 3 objects | | |
|---|---|---|---|---|---|---|---|---|---|---|---|---|
| STORM* | 114 it/s | 8.1 it/s | 3.67 h | - | | | - | | | - | | |
| OC-STORM (obj module only) | | - | | 32 it/s | 8.8 it/s | 4.02 h | 32 it/s | 8.5 it/s | 4.14 h | 31 it/s | 7.8 it/s | 4.46 h |
| OC-STORM (both modules) | | - | | 28 it/s | 5.9 it/s | 5.70 h | 27 it/s | 5.5 it/s | 6.08 h | 27 it/s | 5.3 it/s | 6.27 h |

## G Details for the Use of Cutie

### G.1 Number of Annotations and Input Resolutions

To prompt Cutie, we use 6 annotation masks per Atari game and 12 per Hollow Knight boss. One potential critique is that few-shot annotation requires prior knowledge of the environment, which may seem unsuitable for general agent learning. However, we view this process as akin to informing the agent of certain task rules. While rewards can reflect task rules, they are often too sparse to facilitate an understanding of complex environments. Just as humans may initially struggle to understand how to play a game without being told the rules, there is no reason not to inform agents of key objects. Therefore, we believe this pipeline holds practical value in many cases.

For Atari, we upscale the observation from $210 \times 160$ to $420 \times 320$. This upscaling aids in the identification of small objects in Atari games, such as the ball in Pong and Breakout. For each game, we hand annotate 6 masks.

For Hollow Knight, we resize the observation's shorter side to 480p while maintaining the aspect ratio before inputting it into the Cutie. For each game, we hand annotate 12 masks.

### G.2 Modifications for Integration with STORM

We make no modifications to the official implementation, except for caching and copying internal variables.

The only special process involves setting the object feature vector to 0 when Cutie loses track of the object. Cutie uses an attention guidance mask within its object transformer, which restricts which visual features the object feature can attend to. This mask is trained as part of an auxiliary segmentation task. When the attention guidance mask is set to all 1s (0 allows attention and 1 rejects it), indicating that Cutie cannot find strong evidence of the object's presence in the scene, the transformer theoretically should reject all attention from the visual features.

However, in this situation, Cutie inverts the mask, allowing the object feature to attend to all visual features in an attempt to search for the object in the scene. As a result, the attention becomes scattered across the observation space, leading to unpredictable output for the object feature. This unpredictable behaviour complicates learning in the world model.

To address this, we set the object feature vector to 0 when the attention guidance mask is entirely 1. This informs the world model that the object feature is missing, rather than reflecting a random state.

# H   Hyperparameters

Table 7: Hyperparameters for both Atari and Hollow Knight. The life loss information configuration aligns with the setup used in EfficientZero (Ye et al., 2021). Regarding data sampling, each time we sample $B_1$ trajectories of length $T$ for world model training, and sample $B_2$ trajectories of length $C$ for starting the imagination process. The train ratio is defined as the number of gradient steps over the number of environment steps.

| Hyperparameter | Symbol | Value |
|---|---|---|
| Transformer layers | $K$ | 2 |
| Transformer feature dimension | $D$ | 256 |
| Transformer heads | - | 4 |
| Dropout probability | $p$ | 0.1 |
| World model training batch size | $B_1$ | 32 |
| World model training batch length | $T$ | 32 |
| Imagination batch size | $B_2$ | 512 |
| Imagination context length | $C$ | 4 |
| Imagination horizon | $L$ | 16 |
| Train ratio | - | 1 |
| Environment context length | - | 16 |
| Gamma | $\gamma$ | 0.985 |
| Lambda | $\lambda$ | 0.95 |
| Entropy coefficiency | $\eta$ | $1 \times 10^{-3}$ |
| Critic EMA decay | $\sigma$ | 0.98 |
| Optimizer | - | Adam (Kingma & Ba, 2015) |
| Activation functions | - | SiLU (Elfwing et al., 2018) |
| World model learning rate | - | $1.0 \times 10^{-4}$ |
| World model gradient clipping | - | 1000 |
| Actor-critic learning rate | - | $3.0 \times 10^{-5}$ |
| Actor-critic gradient clipping | - | 100 |
| Gray scale input | - | False |
| Frame stacking | - | False |
| Atari frame skipping | - | 4 (max over last 2 frames) |
| Hollow Knight FPS | - | 9 |
| Use of life information | - | True |

# I   Illustration of Limitations

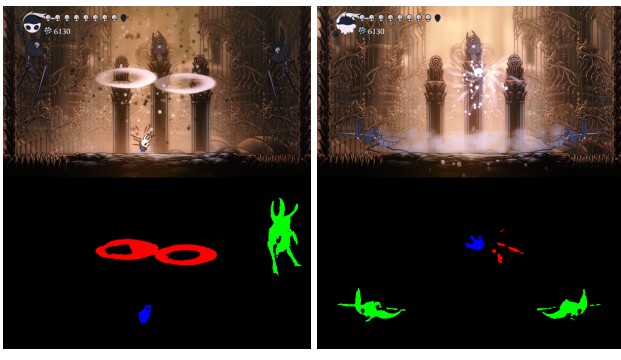

(a) Miss one lord.          (b) Succesfully identify two lords.

Figure 15: Sample frame and segmentation masks generated by Cutie from the Hollow Knight Mantis Lords. Cutie may lose track of one of the lords (represented with green masks). This tracking issue is more likely to occur not only in this scenario but also in other environments where duplicated instances are present, compared to scenes with a single instance.

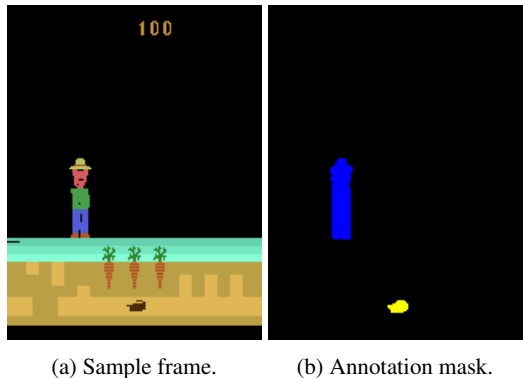

(a) Sample frame.          (b) Annotation mask.

Figure 16: Sample frame and annotation segmentation masks from Atari Gopher. We only specify two objects for Gopher. The tunnel in the ground is challenging to encode as an object given our model structure.

## J    Sample Annotations for Atari Games and Hollow Knight

Figure 18 and 17 present sample frames and annotations used by our method in Atari and Hollow Knight, respectively. For each Atari game, we annotate 6 frames, and for each boss in Hollow Knight, we annotate 12 frames.

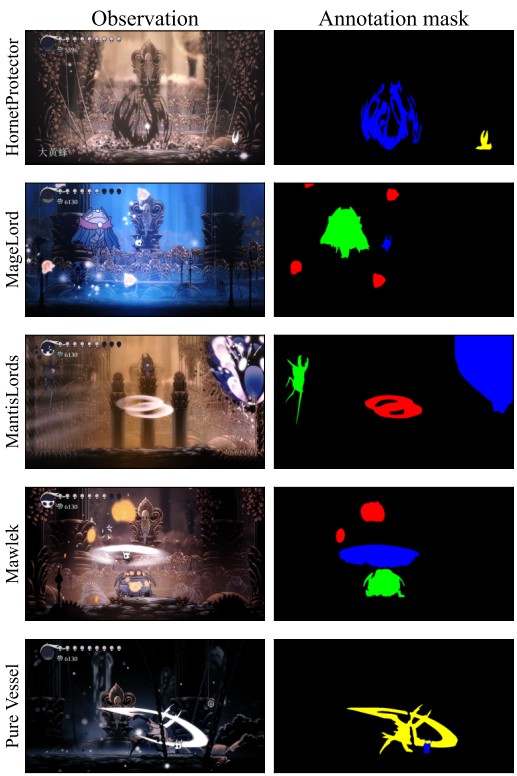

Figure 17: Sample frames and annotation masks for Hollow Knight bosses.

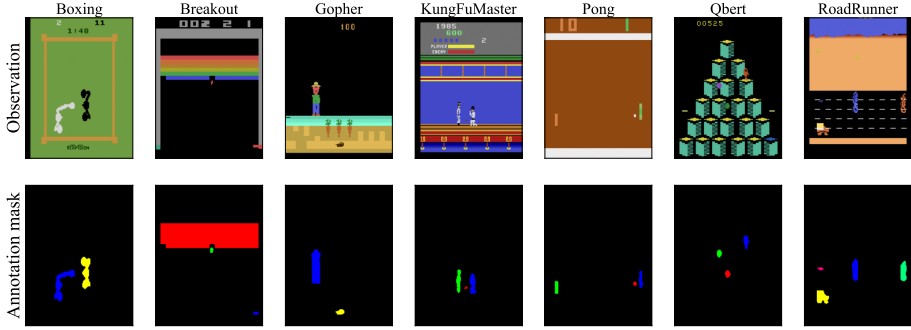

Figure 18: Sample frames and annotation masks for Atari games.