# OpenReview forum: "Objects Matter: Object-Centric World Models Improve Reinforcement Learning in Visually Complex Environments"
_rl-conference.cc/RLC/2025/Workshop/RLVG — RLVG Workshop - RLC 2025_

### Official Review · Reviewer_WFEg · 2025-06-16
**Objects Matter: Object-Centric World Models Improve Reinforcement Learning in Visually Complex Environments**

**Rating:** 3
**Confidence:** 3

**Summary:**

The authors extend algorithm STORM, a model-based algorithm that replaces the RNN of the Dreamer model with a transformer, with an object-centric pipeline. Their approach works by manually annotating important key objects of the tasks and then using a foundational vision model (Cutie) to extract the object features of the task. The output of this foundational model is then concatenated with the vision input of the STORM model. The authors show improvements by using this approach (visual + object inputs) vs just using visual inputs.

**Strengths:**

+ The paper is well written
+ It has a good description of the algorithm and good exploration in the appendix
+ It works well in the complex 3d environment
+ The analysis section is a plus

**Weaknesses:**

+ A quantitative analysis on 5.1 would be better.
+ Lack of confidence intervals in the results.
+ "Figure 4 shows that the vector-based representation generally results in stronger performance than the mask-based representation. " I disagree with this statement, as is discussible, both masks and vector have 2 tasks where one is significantly better than the other, and the other tasks are not statistically significant.

**Best Paper Nomination:**

No

**Claims:**

Yes, overall using OC+vision (proposed method) seems to work better than the baselines (vision only)

**Suggestions:**

I wonder what is the impact of increasing the size of the compressed latent of the object-centric approach. An ablation might be interesting


Aren't there better established benchmarks with complex 3d environments? Like Minecraft? I personally think I would like to see a better analysis on Meta-World than Hollow Knight. But more of a personal opinion.

---

### Official Review · Reviewer_Gfmu · 2025-06-17
**Good paper**

**Rating:** 4
**Confidence:** 4

**Summary:**

The paper proposes an extension of the model-based RL method STORM, called Object-Centric-STORM.

The motivation for this work is clearly stated: Many model-based RL methods rely on auto-encoders trained via L2-objectives, which makes these models focus on large, easy-to-reconstruct image patches, typically found in the background, which are often irrelevant for optimal decision making. The paper proposes to augment model-based RL with object-centric computer vision approaches, to facilitate representations focused on key-objects related to decision making.

The paper explains all components of the proposed method well and provides many supporting figures, which makes the paper easy to follow.

The proposed method is evaluated on Atari100k and Hollow Knight. The main results are that OC-STORM outperforms STORM on Atari games that can be categorized as "featuring objects", and on Hollow Knight boss fights, which also features "object-like entities" (the player and the boss). Across the board, OC-STORM matches and slightly exceeds other baselines like DreamerV3 or DIAMOND. A number of additional interesting results are presented in the appendix.

While there are additional experiments that I would find personally interesting to see, the paper already provides a lot of interesting results, and the evaluation procedure is mostly sounds. The paper fits the topic of the workshop and should, therefore, be accepted.

**Strengths:**

The paper is well-written, easy to follow, provides many supportive figures, and conducts experiments and analysis that are relevant to the hypothesis. The results are promising. The paper features many additional explanations, related work discussion, details, and experiments in the appendix, which also benefit understanding and quality of the paper.

**Weaknesses:**

I don't see strong weaknesses in this paper. The motivation is clear, the experimental methodology is sound, but could (of course) be extended, see suggestions.

The Atari100k benchmark is adequate, but fails to capture the asymptotic performance of the proposed method (in relation to the baselines).

For some reason, the results in tables 1, 2, 3 report only the mean and median, the standard deviation over seeds should be reported.

The contribution is justified, although only incremental, since the paper mainly proposes to add object-centric representations to STORM. The object-centric feature extract, cutie, is used out of the box.

The selection of annotations is not discussed. How are frames for annotation and processing with cutie selected? How robust is cutie when the target objects undergo strong visual transformation (e.g. a second boss phase where the appearance of the boss changes drastically)?

**Best Paper Nomination:**

No

**Claims:**

The main claim is that object-centric representations can help model-based RL by providing the policy with features that better represent objects relevant to decision making, in relation to standard, end-to-end baselines that employ L2 reconstruction loss. Clear evidence for this claim is presented in Table 2, Table 3, Figure 3, Figure 4, and Figure 6.

**Suggestions:**

In addition to the results already presented in the paper, I would be curious about the following additional aspects:

Scalability w.r.t the number of objects K, since K seems to be rather small (2 - 4 in Table 4.) It would be interesting to see the performance of the method when it has to keep track of a larger number of objects. How does the complexity of the model scale in this case? How does the method behave if the selected number K is larger or smaller than the number of relevant objects in the environment?

Scalability w.r.t the number of annotations. Appendix F.3 compares 1 annotation and 6 annotations, if I understand it correctly. Of course, there is a trade-off between the number of annotations and the compute requirements of the method. But it would be interesting to see if the performance of the method can be further improved by showing more annotations of the relevant objects.

How does the method perform asymptotically? What is the methods performance, in relation to the baselines, if we allow 1M samples on Atari?

---

### Decision · Program_Chairs · 2025-06-19

**Decision:**

Accept

**Comment:**

This paper introduces OC-STORM, an object-centric model-based reinforcement learning pipeline designed to improve sample efficiency in visually complex environments by focusing on capturing relevant elements. OC-STORM combines segmentation masks, pre-trained vision foundation models, and a world model for more efficient policy learning. The work well-writtena and detailed, and presents strong performance on Atari games and Hollow Knight, showcasing its potential for real-world application in complex visual tasks. The reviewer highlights the absence of confidence intervals in the results and suggests refining the claims to better reflect the statistical significance of the reported values. We strongly encourage the authors to address these points in the camera-ready version.